# Human protein synthesis requires aminoacyl-tRNA pivoting during proofreading

Divya Sapkota [1,2], Karissa Y. Sanbonmatsu [3,4] & Dylan Girodat [1,5] ✉

Rigorous studies have characterized the aa-tRNA selection mechanism in bacteria, which is essential for maintaining translational fidelity. Recent investigations have identified critical distinctions in humans, such as the requirement of subunit rolling and a tenfold slower proofreading step. Although these studies captured key intermediates involved in tRNA selection, they did not elucidate the transitions of aa-tRNA between intermediates. Through diverse structure-based simulations, we simulated 1856 aa-tRNA accommodation events into the human ribosomal A site. Here we show the requirement for a distinct ~30° pivoting of aa-tRNA about the anticodon stem within the accommodation corridor. This pivoting is crucial for navigating the crowded accommodation corridor, which becomes more constrained due to subunit rolling. Subunit rolling-dependent crowding increases the steric contributions of the accommodation corridor during aa-tRNA accommodation, consistent with the tenfold reduction in the rate of proofreading. Furthermore, we show that eEF1A interacts with the accommodating aa-tRNA through conserved basic residues, limiting premature aa-tRNA dissociation from the A site. These findings provide a structural description of the human aa-tRNA selection process and demonstrate that the aa-tRNA alignment relative to the ribosomal catalytic sites is a critical determinant of translational fidelity.

Protein synthesis is dependent on megadalton ribonucleoprotein complexes known as ribosomes, which decode genetic information into polypeptide sequences. Ribosomes decode genetic information - in the form of messenger RNA (mRNA) - three nucleotides at a time (codon), using aminoacyl-tRNA (aa-tRNA) as a substrate. During this process the mRNA forms three Watson-Crick base pairs with the aa-tRNA (codon-anticodon interactions) to select which amino acid to incorporate into the peptide. Maintaining the fidelity of translation during decoding is essential to the integrity of the proteome and has been evolutionarily fine-tuned to an error rate of $10^{-3}$ to $10^{-4}$ or $10^{-4}$ to $10^{-5}$ amino acids are miscoded per codon in bacteria and eukaryotes, respectively[1–10]. The importance of maintaining fidelity during decoding is exemplified by several antibiotics, including aminoglycosides, that disrupt this process in bacteria[11–13] and by the fact that perturbation of translational fidelity can lead to protein aggregates, which have been associated with neurodegenerative diseases[14]. Furthermore, in humans, recent therapies have been designed to target tRNA selection as a strategy for treating cancer, viral infections, and certain monogenic disorders[15–17].

[1]Department of Chemistry and Biochemistry, University of Arkansas, Fayetteville, AR, USA. [2]Cellular and Molecular Biology Program, University of Arkansas, Fayetteville, AR, USA. [3]Theoretical Biology and Biophysics, Theoretical Division, Los Alamos National Laboratory, Los Alamos, NM, USA. [4]New Mexico Consortium, Los Alamos, NM, USA. [5]Alberta RNA Research and Training Institute, Department of Chemistry and Biochemistry, University of Lethbridge, Lethbridge, AB, Canada. ✉e-mail: dylan.girodat@uleth.ca

mRNA decoding and aa-tRNA selection are well studied in bacteria where it occurs through a two-step process involving initial selection and kinetic proofreading, which are separated by irreversible GTP hydrolysis[18,19]. This mechanism has been extensively biochemically characterized and structurally validated through various methodologies including single molecule Fluorescence (Förster) Resonance Energy Transfer (smFRET)[20–30], cryo-electron microscopy (cryo-EM)[31–39], X-ray crystallography[25,29,40–59], and pre-steady state kinetics[60–63]. Initial selection involves Elongation factor Tu (EF-Tu, or eEF1A in eukaryotes) binding to the small subunit (SSU) of the ribosome as a ternary complex composed of EF-Tu, guanosine triphosphate (GTP), and aa-tRNA (Initial Binding complex−IB). Formation of Watson-Crick base-pairs between the mRNA and aa-tRNA follow, bending the tRNA into the A/T position (Codon Recognition complex− CR). The closing of the SSU shoulder facilitates docking of EF-Tu onto the Sarcin-Ricin loop (SRL) of the large subunit (LSU) (GTPase Activated complex−GA)[28,58,61,62,64,65]. GTP hydrolysis separates initial selection from proofreading where the release of inorganic phosphate and conformational rearrangement of EF-Tu allows for the aa-tRNA to begin transition through the accommodation corridor to enter the A site of the ribosome (Accommodated complex−AC)[61,62,66]. Proofreading during aa-tRNA movement into the A site, through reversible fluctuations[30], promotes opportunities for EF-Tu to reengage the aa-tRNA[23] or to reject the aa-tRNA if it is non-cognate. These intermediate conformations of eEF1A and aa-tRNA in complex with eukaryotic ribosomes have also been identified by smFRET and cryo-EM investigations[67–69]. Although these intermediates have been identified, the tRNA selection process is less understood in eukaryotes.

Human aa-tRNA selection has been determined to be tenfold slower than in bacteria and is rate-limited by proofreading[68]. These findings, showing that aa-tRNA selection in humans is kinetically different, suggest a distinct barrier for aa-tRNA selection during proofreading. To achieve the accommodated position, additional movements of the tRNA perpendicular to the intersubunit space towards the LSU were observed in humans, accompanied by subunit rolling[68–70]. Subunit rolling was observed to occur between the GA and AC complexes and led to the closing of the A site and opening of the E site. These additional vectors of movement indicate that tRNA trajectories into the A site of the ribosome in humans diverge from those observed in bacteria. These insights capture critical intermediates of tRNA selection but do not resolve transitions of aa-tRNA from one intermediate to another, nor do they identify the barrier leading to tenfold reduced rate of tRNA selection.

A powerful tool for describing protein folding[71], oligomerization[72], tRNA selection and tRNA translocation have been structure-based molecular simulations[30,73]. These simulations have described that aa-tRNA must navigate the accommodation corridor through reversible fluctuations as it enters the A site of the ribosome[30,74,75]. Moreover, they have described the roles of EF-Tu[23,76–79], the A-site finger (helix 38 of the 23S rRNA)[80], L11 flexibility[81], tRNA selection[82], and tRNA diffusion[83], as well as ribosome-stimulated EF-Tu GTP hydrolysis[84–89]. In this work we employ structure-based simulations to compare the structural dynamics of tRNA selection in *H. sapiens* and in *E. coli* to determine the structural distinctions between these systems during proofreading. Here we show that intersubunit rolling occurs early during proofreading and that there is a distinct pivoting of the aa-tRNA during human aa-tRNA selection. This pivoting is required due to the increased contact surface between the tRNA and the accommodation corridor caused by subunit rolling. Furthermore, we identified that domain III (DIII) of eEF1A interacts with the elbow domain of the accommodating aa-tRNA through conserved basic amino acids, elucidating how it contributes to aa-tRNA selection. The requirement of tRNA pivoting and eEF1A interactions with the aa-tRNA provides a structural framework for

understanding why tRNA selection is tenfold slower in *H. sapiens* compared to *E. coli* and supports previously proposed studies suggesting that tRNA alignment is essential to selection.

## Results

### Simulations of aa-tRNA proofreading in humans

Accommodation of aa-tRNA into the A site of the human ribosome was simulated using structure-based simulations, which were previously used to simulate bacterial tRNA selection[30,75–77,80,90]. Simulations were initiated in the post-GTP hydrolysis state of eEF1A in complex with aa-tRNA, GDP, and the ribosomal GTPase-activating center (Fig. 1A, B). During simulations, the tRNA approached the A site of the ribosome where the attractive term for native contacts of the accommodated position were defined by potential 1 (attractive 6−12 potential) or potential 2 (attractive Gaussian potential) and non-native contacts were given a repulsive term (Fig. 1C, D; "Methods"). Native contacts are defined as atom pairs within 4.5 Å of each other in the ribosome when the aa-tRNA is in the accommodated A/A conformation, while non-native contacts refer to interactions not present in this conformation. Similarly, coordinates of eEF1A in the open conformation were defined by native contacts, enabling the protein to undergo conformational change from a compacted to extended conformation during release and accommodation of the aa-tRNA. Simulations with potential 1 allowed us to capture the reversible fluctuations of the aa-tRNA as it traversed the accommodation corridor of the ribosome. In these simulations the aa-tRNA fluctuates between elbow-accommodated conformations (EA-1 and EA-2) that were previously identified in

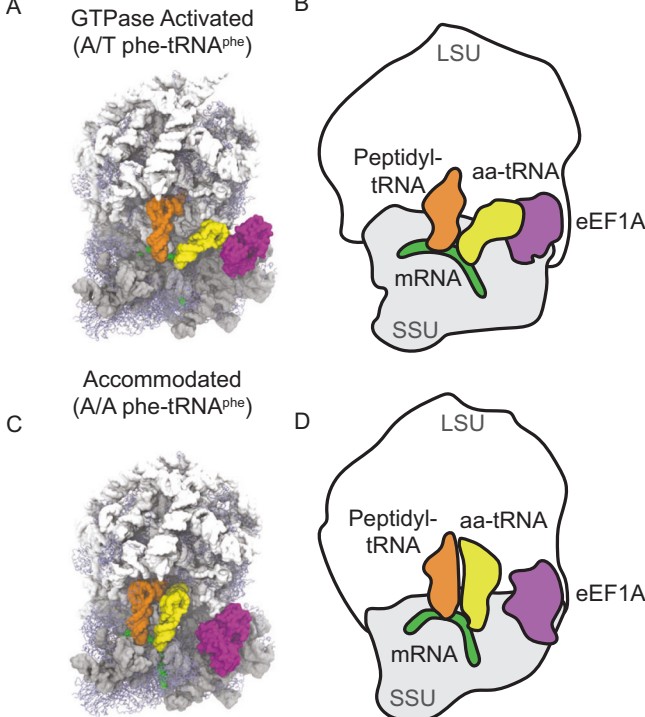

**Fig. 1 | Conformations of ribosomes during eukaryotic aa-tRNA accommodation. A** All-atom structural representation of the pre-accommodated GTPase activated state with aa-tRNA in the A/T position. **B** Simplified model of the GTPase activated state with aa-tRNA in the A/T position. **C** All-atom structural representation of the accommodated state with aa-tRNA in the A/A position. **D** Simplified model of the accommodated state with aa-tRNA in the A/A position. In these models aa-tRNA (yellow), mRNA (green), peptidyl-tRNA (orange), 28S, 5S, 5.8S rRNA (white), 18S (gray), ribosomal proteins (blue) and eEF1A (purple) are represented.

bacteria by cryo-EM during aa-tRNA selection[37]. Simulations with potential 2 enabled us to identify early- and late-stage events during aa-tRNA accommodation as the 3′-CCA end achieved the fully accommodated conformation and facilitated comparisons between bacterial and eukaryotic aa-tRNA accommodation (Supplementary Methods, Supplementary Movies 1, 2). Although simulations with potential 2 enabled us to characterize early and late stage-events of aa-tRNA accommodation, each trajectory only captured 1 accommodation event. An accommodation event was considered when the distance between the aa-tRNA and peptidyl-tRNA elbow domains ($R_{elbow}$) reached a distance of <32.5 Å. The convergence of both types of simulations were defined by the typical measurement of backbone RMSD of the ribosome or aa-tRNA in addition to the pointwise RMSD between successive free energy landscapes $R_{elbow}$ and of $\theta_{tRNA}$, (Supplementary Figs. 1, 2). These simulations were determined to be converged when RMSD or the Conv(t) value approaches a plateau as described previously[90,91]. Simulations using potential 1 converged at 50–500 μs and simulations using potential 2 converged at 5–20 μs, depending on if RMSD or Conv(t) was used. From these simulations, we identified 1769 elbow accommodation events with potential 1 and 87 with potential 2. In simulations using potential 2 we observed aa-tRNA dissociation prior to accommodation in 13 simulations, representing rejected aa-tRNA. Within these simulations 1 reduced unit of time ($\tau_{ru}$) was approximated to be 1 ns, in line with previous investigations using structure-based simulations to quantify tRNA accommodation[30,75,90].

To achieve reversible fluctuations of the aa-tRNA in the simulations with potential 1 we scaled the weight of the contacts for the tRNA in the A/A position in the ribosome. By titrating the weight of the native contacts for the A/A position of the tRNA we were able to fine-tune our simulations to be evenly distributed between the two elbow accommodated positions (EA-1 and EA-2) in the simulations at a contact weight of 0.13 (Supplementary Fig. 3). Similarly, the contacts for the A/A position in potential 2 simulations were scaled by 0.4 to enable fluctuations of the aa-tRNA during accommodation, as previously performed in bacterial systems[90]. A rescaling of 0.4 enables the tRNA to accommodate into the A-site through reversible fluctuations, reflecting tRNA trajectories seen in smFRET studies[28,30]. The interactions between the mRNA and tRNA were scaled by a factor of 0.8 to compare to bacterial studies where it was rescaled by this value to reflect tRNA rejection frequencies[90].

## Human aa-tRNA accommodation requires movement of the elbow domain and 3′-CCA end

Bacterial aa-tRNA accommodation is known to occur through an elbow-first mechanism where the elbow domain of the aa-tRNA (D Loop and T Loop, nucleotides 14–21 and 54–60, respectively) enters the A site of the ribosome prior to the 3′-CCA end. In humans, we measured the trajectory of the aa-tRNA elbow domain into the ribosomal A site by tracking the distance between the O3′ atoms of U60 and U8 on the aa-tRNA and peptidyl-tRNA, respectively ($R_{elbow}$) (Fig. 2A). Additionally, we monitored the accommodation of the 3′CCA end of the tRNA, conjugated to either the nascent polypeptide chain or amino acid, by measuring the distance between the A76 residues of both the peptidyl and aa-tRNA ($R_{CCA}$) (Fig. 2B). We observed the tRNA elbow accommodation as $R_{elbow}$ decreased from ~57 Å to ~32 Å, a distance change of ~25 Å (Fig. 2C; Supplementary Fig. 4). Similarly, the 3′-CCA end of the aa-tRNA moved into the A site of the ribosome as it goes from ~80 Å to ~8 Å, a 72 Å change (Fig. 2D; Supplementary Fig. 5). The $R_{CCA}$ measurement shows a clear intermediate position at 20–40 Å, an intermediate position previously identified in bacteria where the tRNA contacts regions of the accommodation corridor such as H71 and H89 of the 23S rRNA (Fig. 2D). We observed that this intermediate for human aa-tRNA accommodation contacts the same

regions of the ribosome in the intermediate position (Supplementary Fig. 6).

At longer timescales (up to 1.5 ms) of simulation using potential 1, we observe the reversible fluctuation of the aa-tRNA into the A-site of the ribosome, reflecting repetitive accommodation attempts. This is captured in both the $R_{elbow}$ value of the simulations and in the $R_{CCA}$ value of the simulations (Fig. 2E, F; Supplementary Figs. 7, 8). In these trajectories, we see that the movement of the elbow domain and of the 3′CCA region are highly coupled meaning that as one approaches the A/A or A/T position, the other follows. By using the number of time-steps required for each accommodation event in our simulations we could estimate the barrier height during transition from the A/T position to the transition state using previously determined rates of accommodation (Supplementary Methods). From these measurements we estimate that the barrier height of *E. coli* accommodation is 9.3–10.4 $k_B T$, similar to previously determined values, while in humans the barrier is 11–13.4 $k_B T$[75].

## Human aa-tRNA accommodates through a distinct 3′-CCA first approach

From simulations of the reversible accommodation of aa-tRNA into the A site of the ribosome, we generated a Boltzmann-weighted approximate free energy landscape comparing $R_{elbow}$ and $R_{CCA}$. In this landscape, we observe two dominant positions, one at a $R_{elbow}$ and $R_{CCA}$ value of ~42 Å and ~60 Å, respectively denoted elbow accommodated-1 (EA-1) and another at ~32 Å and ~25 Å, respectively denoted elbow accommodated-2 (EA-2) (Fig. 3A). The nomenclature of EA-1 and EA-2

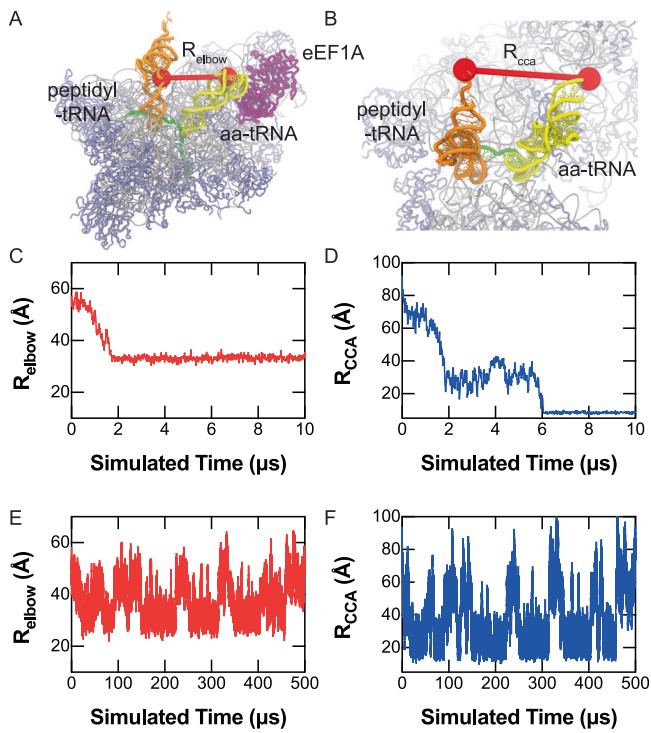

**Fig. 2 | Dynamics of human aa-tRNA accommodation. A** Distance measured between O3′ atoms of U60 and U8 of the aa-tRNA and peptidyl-tRNA, respectively, to characterize the elbow accommodation dynamics of the aa-tRNA ($R_{elbow}$). **B** Distance measurement between the O3′ atoms of A76 of the aa-tRNA and peptidyl tRNA during accommodation into the A site of the ribosome ($R_{CCA}$). **C** Representative distance of $R_{elbow}$ during simulation using potential 2. **D** Representative distance of $R_{CCA}$ during a simulation using potential 2. **E** Representative distance of $R_{elbow}$ during a simulation using potential 1, showing reversible fluctuations. **F** Representative distance of $R_{CCA}$ during a simulation using potential 1, showing reversible fluctuations. Source data are provided as a Source Data file.

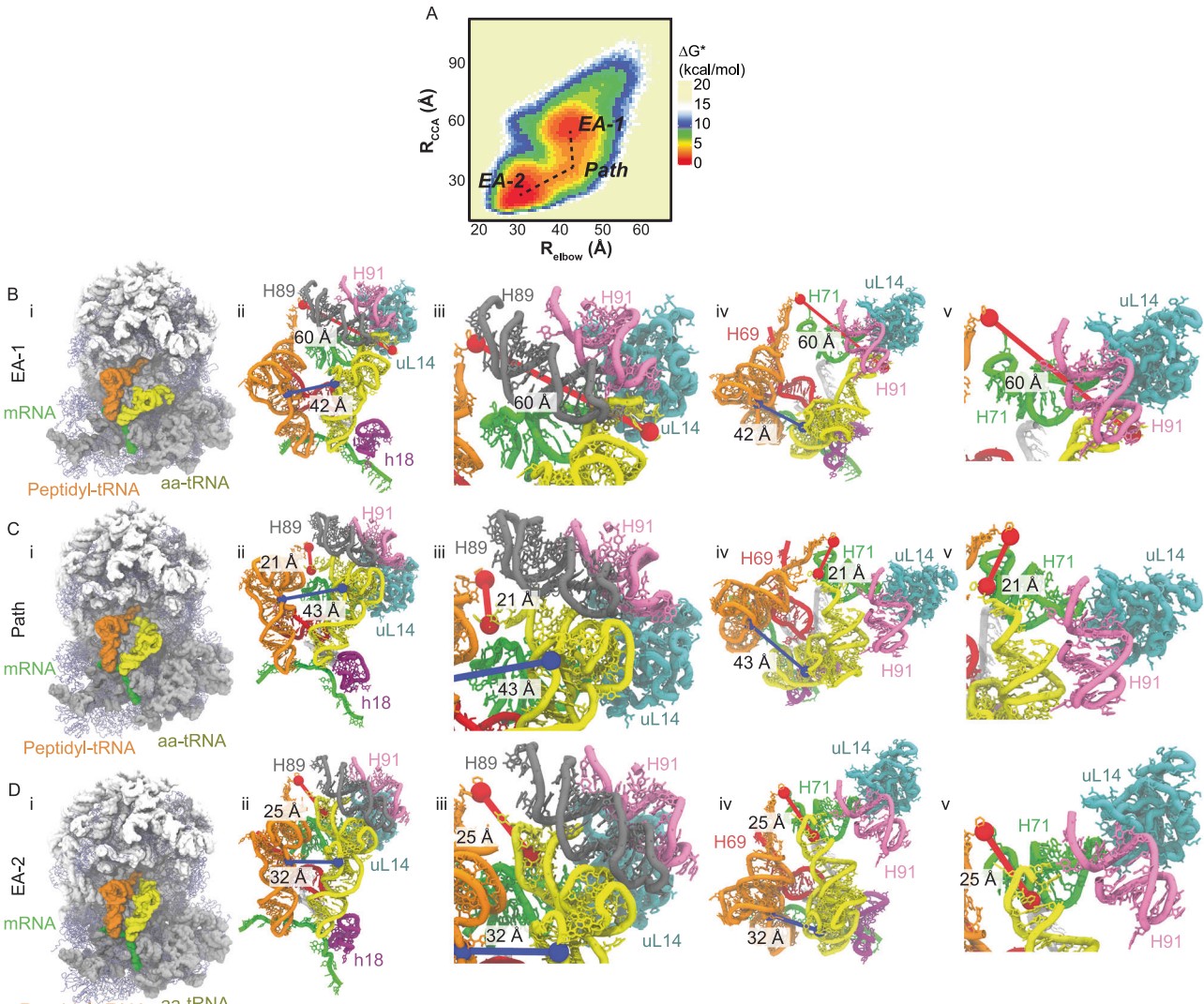

**Fig. 3 | Human aa-tRNA accommodation occurs through a 3′-CCA end dependent pathway. A** Approximate free energy ($\Delta G^*$) landscape of tRNA accommodation as measured by comparison of the $R_{elbow}$ and $R_{CCA}$ distance measurements from simulations using potential 1. **B** Representative structure of EA-1. Complete 80S structure of EA-1 (**i**). Representation of the accommodation corridor of the ribosome with aa-tRNA engaging H89 in EA-1 (**ii**). Zoom in of accommodation corridor with aa-tRNA engaging H89 in EA-1 (**iii**). Representation of the accommodation corridor in EA-1 from the LSU perspective (**iv**). Zoom in of the 3′ CCA end of the aa-tRNA interacting with H71 in EA-1 (**v**). **C** Representative structure of the accommodation corridor during transition path from EA-1 to EA-2 (Path). Complete 80S structure of Path (**i**). Representation of the accommodation corridor of the ribosome with aa-tRNA engaging H71 in Path (**ii**). Zoom in of the Path position with the aa-tRNA interacting with H89 (**iii**). Representation of the accommodation corridor in Path from the LSU perspective (**iv**). Zoom in of the 3′ CCA end of the aa-tRNA interacting with H71 in Path (**v**). **D** Representative structure of EA-2. Complete 80S structure of EA-2 (**i**). Representation of the accommodation corridor of the ribosome with aa-tRNA engaging H71 after passing H89 (**ii**). Zoom in of the EA-2 position with the aa-tRNA interacting with H89 (**iii**). Representation of the accommodation corridor in EA-2 from the LSU perspective (**iv**). Zoom in of the 3′ CCA end of the aa-tRNA interacting with H71 in EA-2 (**v**). In these models $R_{elbow}$ highlighted in blue and $R_{CCA}$ highlighted in red, while the rRNA (white and gray), ribosomal proteins (blue), peptidyl-tRNA (orange), aa-tRNA (yellow), H89 (gray), H90 (mauve), H71 (green), h18 (purple), H44 (red), and mRNA (green) are represented. Source data are provided as a Source Data file.

are used instead of A/T and A/A as the aa-tRNA in these positions differ from the $R_{elbow}$ and $R_{CCA}$ of the A/T and A/A from resolved structures. In the A/T position the $R_{elbow}$ and $R_{CCA}$ are 57 Å and 79 Å, respectively, and in the A/A state they are 32 Å and 8 Å, respectively. The EA-1 and EA-2 positions reflect the elbow-accommodation aa-tRNA position that has been observed in *E. coli* previously[37], which are on-path towards the A/A tRNA position. Both the EA-1 and EA-2 positions were observed in our simulations of bacterial accommodation (Supplementary Fig. 9). The $R_{elbow}$ and $R_{CCA}$ values of the EA-1 position in *E.coli* structures are ~36 Å and ~47 Å and the EA-2 position are ~35 Å and 22 Å, respectively. In comparing bacterial and human elbow accommodated positions, EA-1 has distinct $R_{elbow}$ and $R_{CCA}$ distances compared to humans but the EA-2 position is similar between species.

EA-1 reflects an aa-tRNA arrangement whereby the aa-tRNA elbow domain and the 3′-CCA end have initiated movement into the ribosome and have contacted H89, which sterically restricts the tRNA movement (Fig. 3B). In EA-2 the aa-tRNA has passed H89 and is approaching final A/A position, however, the 3′-CCA contacts H71, posing as a steric barrier for tRNA movement (Fig. 3C). An observable difference between *H. sapiens* and *E. coli* tRNA accommodation is that the tRNA 3′-CCA end moves towards the A-site prior to the elbow domain in the transition between EA-1 and EA-2 (Fig. 3A). This pathway (Path) appears to be distinct in eukaryotic tRNA selection and involves the 3′-CCA end of the tRNA approaching the H71 barrier prior to overcoming the H89 barrier (Fig. 3D). This indicates that the steric barrier imposed by

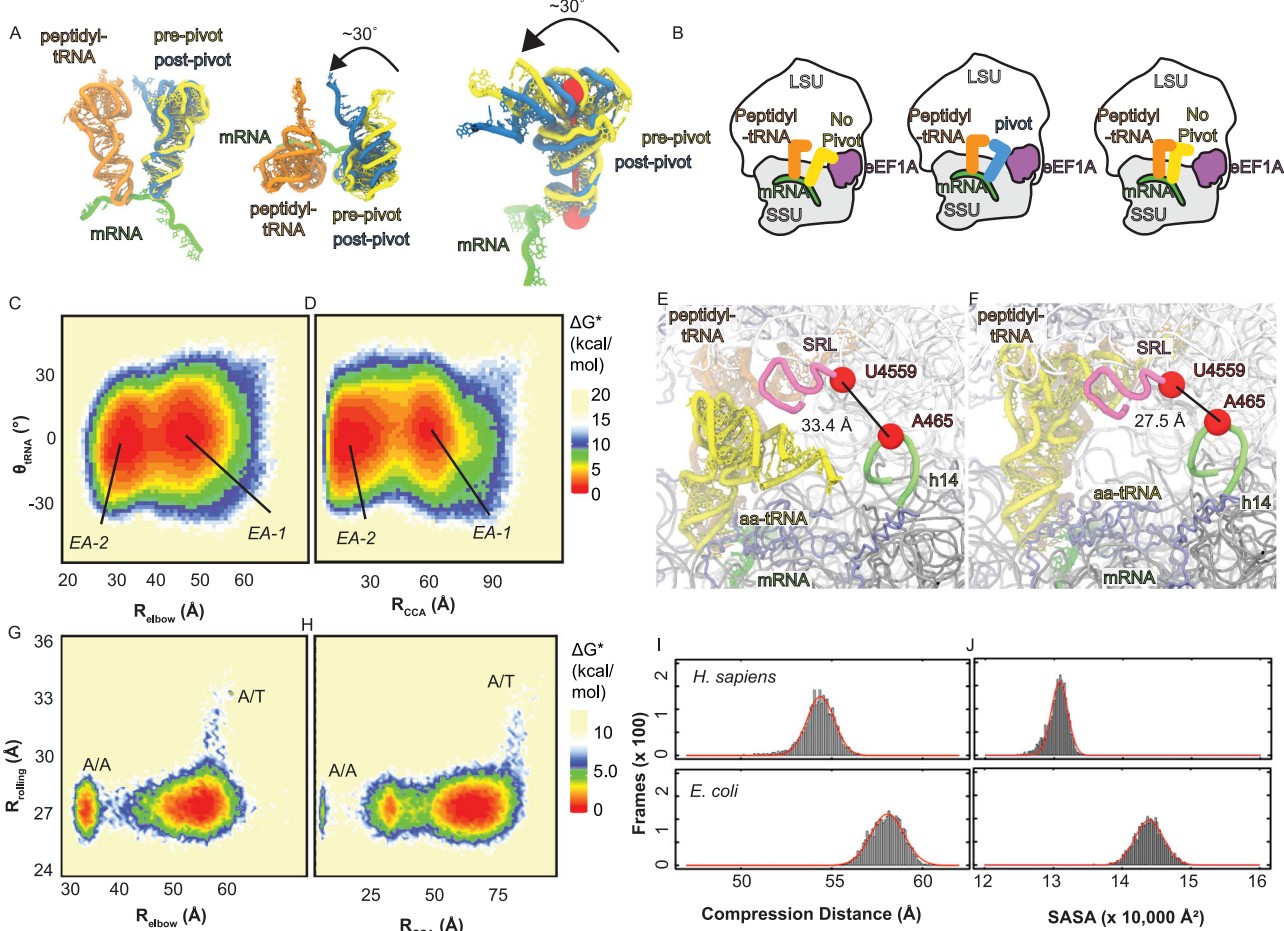

**Fig. 4 | Human aa-tRNA pivoting during accommodation into the ribosomal A site. A** Representation of aa-tRNA pivoting ($\theta_{tRNA}$) during accommodation into the A site. Perspective from the central protuberance (left), LSU (middle), and P site (right) with peptidyl-tRNA (orange), mRNA (green), pre-pivot aa-tRNA (yellow), and post-pivot aa-tRNA (blue). $\theta_{tRNA}$ is the angle of the tRNA as it moves into the ribosomal A site. **B** Representations of the aa-tRNA pivoting into the ribosomal A site. aa-tRNA begins accommodation in the bent non-pivoted position (left), the tRNA pivots to accommodate through a 3'CCA-end first mechanism (middle), then the tRNA returns to a non-pivoted position once the elbow of the tRNA enters the A site. **C** Approximate free energy landscape of human aa-tRNA accommodation into the ribosomal A site generated by comparing the $R_{elbow}$ distance and $\theta_{tRNA}$ angle, from potential 1 simulations. **D** Approximate free energy landscape of human tRNA accommodation into the ribosomal A site generated by comparing the $R_{CCA}$

distance and $\theta_{tRNA}$ angle, from potential 1 simulations. **E** Representation of the open A site of the ribosome in the GA complex, prior to accommodation. **F** Representation of the closed A site of the ribosome in the AC complex, after accommodation. **G** Approximate free energy landscape of human ribosome rolling during tRNA accommodation generated by comparing $R_{elbow}$ and $R_{rolling}$, from potential 2 simulations. **H** Approximate free energy landscape of human ribosome rolling during tRNA accommodation generated by comparing $R_{CCA}$ and $R_{rolling}$, from potential 2 simulations. **I** Histogram of distances between atoms A35 and C61 of the accommodating *H. sapiens* and *E. coli* aa-tRNA, representing the compression of tRNA during accommodation ($R_{compression}$), from potential 2 simulations. **J** Solvent accessible surface area (SASA) of aa-tRNA during *H. sapiens* and *E. coli* during accommodation, from potential 2 simulations. Source data are provided as a Source Data file.

H89 is larger in humans than it is in bacteria, which could account for the tenfold reduction in the rate of tRNA selection.

## Human tRNA pivots during accommodation into the A site

For aa-tRNA to accommodate into the ribosomal A-site using the 3'-CCA end first pathway that we observe between EA-1 and EA-2, then the tRNA would have to be positioned differently than it is in bacteria. To quantify the different positioning of the tRNA, we measured the change in angle of the tRNA over time ($\theta_{tRNA}$) (Fig. 4A, B; Supplementary Fig. 10A). Specifically, $\theta_{tRNA}$ is the change in angle of a vector perpendicular to the plane of the tRNA generated from atoms O3' of C4, A35, and G56. This angle reflects the pivoting of the tRNA with respect to the codon-anticodon interaction along the acceptor stem and D-loop of the tRNA. In a manner distinct from bacteria, we observed key pivoting of the human aa-tRNA during accommodation into the A site (Supplementary Fig. 10B). This pivoting occurs at the early stages of tRNA accommodation as measured by $\theta_{tRNA}$ and

appears to match the same temporal regime (2–6 μs) as it takes the tRNA to approach the $R_{CCA}$ intermediate position (Fig. 2D). No pivoting was observed in our bacterial simulations (Supplementary Fig. 9B). When comparing $\theta_{tRNA}$ with the $R_{elbow}$ distance on an approximate free energy landscape we observe that as $R_{elbow}$ distances decrease, indicating the tRNA is moving from EA-1 to EA-2, that the angle of the tRNA displays variability of −30 to 30° (Fig. 4C). This pivoting of the aa-tRNA narrows to an energetic minimum of ~0° at a $R_{elbow}$ value of ~38 Å, a distance between EA-1 and EA-2 where the aa-tRNA is in contact with H89 (Fig. 3A, C). Similarly, we compared the $\theta_{tRNA}$ value compared to the $R_{CCA}$ distance measurement. Through this comparison we see a similar trend whereas the tRNA transitions from EA-1 to EA-2, the tRNA goes through an energy minimum of ~0°, although variability in the tRNA angle is available (Fig. 4D). Here, we see that the narrowing of the tRNA angle occurs at an $R_{CCA}$ value of ~35 Å, occurring prior to 3'CCA end contacts H71, $R_{CCA}$ distance ~25 Å (Fig. 2D). This means that the tRNA pivoting in the accommodation corridor is not to surpass the H71

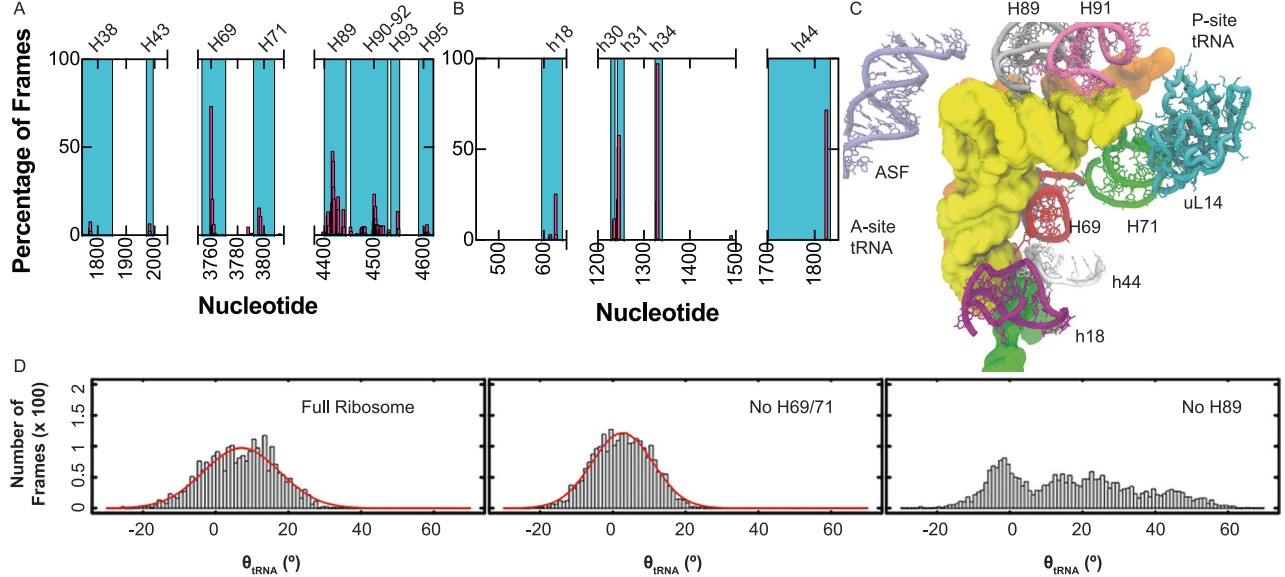

**Fig. 5 | aa-tRNA interactions with the accommodation corridor. A** Regions of the 28S rRNA that are within 4 Å of the aa-tRNA during accommodation simulations of the tRNA into the ribosomal A site. **B** Regions of the 18S rRNA that are within 4 Å of the aa-tRNA during accommodation simulations of the tRNA into the ribosomal A site. **C** Structural representation of rRNA and ribosomal protein uL14 that are in proximity to aa-tRNA during accommodation. **D** Histogram of $\theta_{tRNA}$ during accommodation of tRNA, captured from $N \geq 5$ potential 2 simulations composed of full ribosome or with no H69/71 or no H89. Source data are provided as a Source Data file.

barrier. The pivoting aligns with the timing required for the tRNA to surpass the H89 barrier. This indicates that the tRNA is flexible in the accommodation corridor in eukaryotes, enable the 3'-CCA end to reach the H71 barrier, however, to overcome the H89 barrier, the tRNA has to achieve a $\theta_{tRNA}$ of ~0° to enter the A site.

## The human accommodation corridor is compacted by subunit rolling

As subunit rolling is known to occur during aa-tRNA accommodation between the GA and AC complexes[68–70], we investigated if it contributes to the aa-tRNA pivoting observed in structure-based simulations (Supplementary Fig. 11). To quantify subunit rolling, we measured the distance between the O3' atoms of A465 of h14 and U4559 of the SRL ($R_{rolling}$) (Fig. 4E, F). At the beginning of our simulations, we observed a $R_{rolling}$ distance of 33.4 Å, and at the end of the simulation, this distance narrowed to 27.5 Å, consistent with subunit rolling, leading to a closure of the A site of the ribosome (Fig. 4C, D). We investigated how subunit rolling correlates with the aa-tRNA accommodation to investigate if the closure of the A site could influence the tRNA pivoting. By comparing $R_{rolling}$ with $R_{elbow}$ on an approximate free energy landscape, we identified that subunit rolling occurs prior to elbow accommodation (Fig. 4G). As subunit rolling occurred early in our simulations, we had to perform these comparisons in simulations using potential 2. These landscapes show that $R_{rolling}$ decreases from 33.4 Å to ~27.5 Å prior to $R_{elbow}$ shifts from ~57 Å to 32 Å (Fig. 4G). Similar behavior is observed when comparing $R_{rolling}$ with $R_{CCA}$ (Fig. 4H). We observed that $R_{rolling}$ decreased prior to $R_{CCA}$ shifts from ~80 Å to ~8 Å (Fig. 4H). This suggests that as the aa-tRNA accommodates, subunit rolling has already occurred.

To provide an understanding as to how subunit rolling impacts the accommodating aa-tRNA, we investigated the steric constraints that the accommodation corridor imposes on the aa-tRNA. Initially, we investigated the compression of the tRNA (the distance between residues A35 and C61 of the tRNA) to identify whether the accommodation corridor is narrower (Fig. 4I). This measurement showed that in humans, to enter the A site, the tRNA is compressed by 3.7 Å, as from our Gaussian fit, we identify the tRNA compression to be

54.4 ± 0.8 Å in *H. sapiens* and 58.1 ± 0.9 Å in *E. coli* (Fig. 4I). Similarly, we measured the solvent accessible surface area (SASA) of the tRNA in the accommodation corridor (Fig. 4J). Here, we observe *H. sapiens* tRNA having 13079 ± 121 Å² SASA and *E. coli* tRNA having 14390 ± 206 Å² SASA, indicating that the tRNA is surrounded by more of the ribosome in *H. sapiens* than in *E. coli* (Fig. 4J). Surprisingly, the SASA values did not appear to change much throughout the course of our trajectories, indicating that subunit rolling induced crowding of the accommodating tRNA provides uniform contacts between the tRNA and ribosome or eEF1A throughout accommodation (Supplementary Fig. 12). Together, these data support that the accommodation corridor in humans is more compact than in bacteria due to subunit rolling.

## H89 of the tRNA accommodation corridor influences tRNA pivoting angles

To identify which region of the aa-tRNA accommodation corridor influences the pivoting angles of aa-tRNA during accommodation, we examined all regions of the 28S, 18S, and ribosomal proteins that are within 4 Å of the aa-tRNA during accommodation (Fig. 5A, B; Supplementary Fig. 13). Using this method to identify the ribosomal regions that contact the aa-tRNA, we found that H38, H69, H71, and H89 make the majority of contacts with the tRNA in the 28S rRNA (Fig. 5A). Within the 18S rRNA, h18, h30, h31, h34, and h44 make contact with the aa-tRNA (Fig. 5B). We observed regions such as H69, h30, h31, and h44 making contacts for the majority of the frames, as the aa-tRNA remains in constant contact with these regions throughout the simulation. These ribosomal elements are near the codon-anticodon interactions, which remain relatively stationary during the accommodation of aa-tRNA. Compared to the rRNA, ribosomal proteins made fewer contacts with the aa-tRNA; however, ribosomal proteins uL16, uL14, uL6, and eS30 made the most contacts with the aa-tRNA (Supplementary Fig. 13). These ribosomal regions are consistent with what we observe in simulations as the aa-tRNA moves through the accommodation corridor (Fig. 5C).

To determine which regions of the ribosomal accommodation corridor influence the pivoting of the aa-tRNA we performed

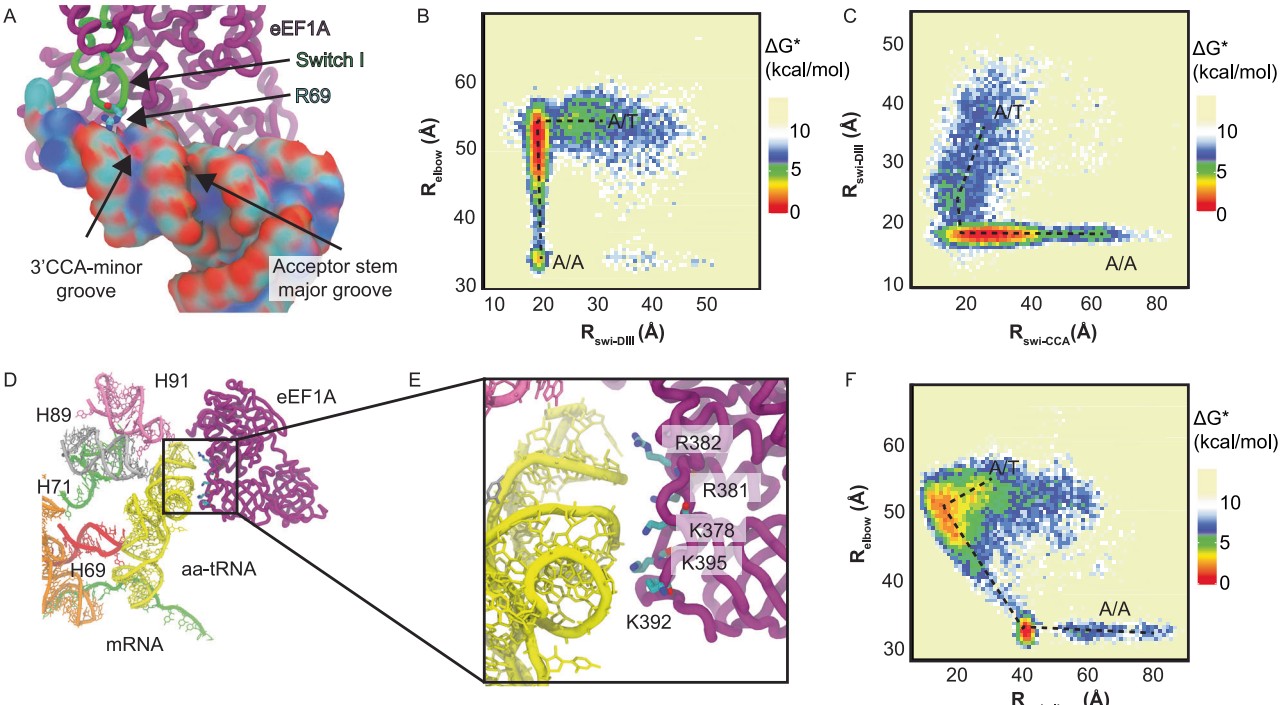

**Fig. 6 | eEF1A interacts with accommodating aa-tRNA through R69 and conserved basic amino acids in Domain III. A** R69 of switch I of eEF1A interacts with the minor groove of the aa-tRNA adjacent to the 3'CCA end. R69 is highlighted interacting with the phosphodiester backbone of the tRNA. **B** Approximate free energy landscape of $R_{elbow}$ in comparison to the distance between eEF1A R69 and A425 of domain III ($R_{swi-DIII}$), generated from potential 2 simulations. This landscape indicates that for aa-tRNA to accommodate, R69 of switch I needs to pass through the 3'CCA-minor groove of the aa-tRNA and dock on domain III. **C** Approximate free energy landscape of $R_{swi-DIII}$ with respect to the distance of R69 and A76 of aa-tRNA, generated from potential 2 simulations. This landscape highlights that only the path where R69 passes by the 3'CCA-minor groove is available. **D** Interactions of eEF1A domain III with in proximity to the accommodating aa-tRNA. **E** Zoomed in image of domain III of eEF1A interacting with the accommodating aa-tRNA. Basic amino acids are highlighted that could interact with the aa-tRNA. **F** Approximate free energy landscape of $R_{elbow}$ distance with respect to K378 to U55 O3' distance, generated from potential 2 simulations. This demonstrates that domain III approaches the accommodating aa-tRNA in all simulations at an $R_{elbow}$ of ~52 Å. Source data are provided as a Source Data file.

structure-based simulations using potential 2 with a virtual removal of H69 and H71 to remove interactions with the acceptor stem and for 3'-CCA accommodation. Secondly, we virtually removed H89 from simulations to identify its influence on aa-tRNA accommodation. H38 or the A-site finger was not removed from simulations as it is structural dynamic and is therefore unlikely to restrict aa-tRNA movements[92]. From these simulations we measured $\theta_{tRNA}$ during the simulation (Fig. 5D). In these simulations, we observed a similar distribution of $\theta_{tRNA}$ for the full ribosome ~−20 to 25°, with a mean of 7.1 ± 10.5° (Fig. 5D). When we deleted H69 and H71, we observed a similar distribution with a mean of 2.6 ± 8.4° (Fig. 5D). This indicates that the pivoting of the aa-tRNA is not dependent on these two helical elements. When we removed H89, we observed an increased distribution of $\theta_{tRNA}$ angles during the simulation, now ranging from −20 to 65° (Fig. 5D). Furthermore, the distribution no longer centered around the mean of 7.1°, indicating that the tRNA is no longer required to restrict the angle distribution to enter the ribosomal A site. This data supports that the aa-tRNA during accommodation must approach a narrow angular distribution to surpass H89 to enter the ribosome and that H89 creates a larger barrier for aa-tRNA entry into the ribosome.

### Switch I of eEF1A interacts with the 3'-CCA end of the accommodating aa-tRNA

Previously, we have reported on the conserved arginine of the switch I element of EF-Tu (R58) that interact with the minor and major grooves of the acceptor stem of the accommodating aa-tRNA before docking onto domain III of eEF1A during early stage accommodation events into the A site of the ribosome[76,90]. To investigate if eEF1A interacts similarly with the accommodating tRNA in humans, we investigated the interactions of R67 and R69 of eEF1A switch I with the acceptor stem of aa-tRNA. Here we observe R69 of switch I interacts with the phosphodiester backbone of the aa-tRNA during early accommodation movements of the aa-tRNA (Fig. 6A). To quantify this, we measured the distance of R69 to and A425 of eEF1a ($R_{swi-DIII}$) to describe the position of switch I relative to domain III and compared it to $R_{elbow}$ (Fig. 6B). This measurement showed that the $R_{swi-DIII}$ distance decreased prior to the $R_{elbow}$ distance, indicating that switch I docks onto domain III of eEF1A prior to accommodation of the aa-tRNA. This requirement is also found in bacteria where switch I needs to interact with and traverse the grooves of the aa-tRNA prior to aa-tRNA accommodation[76]. Secondly, we wanted to identify which grooves of the aa-tRNA does R69 interacts with during accommodation. By comparing $R_{swi-DIII}$ with the distance of R69 to A76 of the 3'-CCA end of the aa-tRNA ($R_{swi-CCA}$), we could identify which grooves of the aa-tRNA R69 passes through (Fig. 6C). Here, we identify only one dominant path that R69 passes through at an $R_{swi-DIII}$ distance of ~22 Å, we see that the $R_{swi-CCA}$ distance is ~15–25 Å. This distance is consistent with R69 passing through the 3'CCA-minor groove of the aa-tRNA and not with it passing through the major groove of the acceptor stem (Fig. 6A). This behavior is distinct from bacteria where R58 can pass through both the 3'CCA-end minor groove and the acceptor stem major groove. As switch I can only traverse the 3'CCA-end minor groove, it indicates that the initial distribution of possible trajectories of the aa-tRNA into the A site is narrower than it is in bacteria.

## Domain III of eEF1A interacts with the aa-tRNA elbow through conserved basic amino acids

To identify any other basic amino acids that make direct contact with aa-tRNA during accommodation, we measured all basic amino acids of eEF1A that are within 4 Å of the accommodating aa-tRNA during the simulation (Supplementary Fig. 14). This analysis showed amino acids that line the aa-tRNA binding site, but also revealed that the basic amino acids K378, K392, K395, R381, and R382, which are part of domain III and distant from aa-tRNA, at the start of the simulation, interact with the accommodating aa-tRNA. From our simulations we observe that as eEF1A domain separation occurs and domain I rotates relative to domain II and III, domain III becomes positioned to interact with the elbow domain of the aa-tRNA (Fig. 6D, E; Supplementary Movie 1). The amino acids K378, K392, K395, R381, and R382 are positioned to interact with the phosphodiester backbone of the accommodating aa-tRNA (Fig. 6D, E). In this position eEF1A would remain bound to the ribosome long enough to facilitate aa-tRNA accommodation by providing a steric barrier for aa-tRNA dissociation from the A site. To determine the likelihood of these interactions being critical to aa-tRNA accommodation, we looked at the conservation of

these basic amino acids (Supplementary Fig. 15). By comparing 154 eEF1A sequences across eukaryotes, we identified the conservation of these basic amino acids (Table 1). From this analysis we find that K378 and K395 are highly conserved at 94.84% and 96.13%, respectively, and are not known to be post-translationally modified, indicating that their basic properties are conserved[93]. Furthermore, if these amino acids are not lysine, they are frequently arginine, indicating that their charge is conserved. The conservation of nucleotides of the tRNA that interact with domain III of eEF1A were also considered, either amongst humans or throughout different domains of life (G1, U51, G52, A64, G65, and U66) (Supplementary Table 2). Although the conservation of these nucleotides varied (0.43–99.31%), all interactions between the tRNA and domain III of eEF1A were mediated by backbone interactions, indicating that the nucleoside identity is not essential for these inter-actions (Fig. 6D,E). Together, these findings indicate that the role these positive amino acids play in aa-tRNA accommodation is conserved.

## Discussion

Here, we identified a pivoting of the aa-tRNA during accommodation into the ribosomal A site of the ribosome, which is not observed in bacteria. The ribosomal accommodation corridor appears to be more crowded in humans, as observed by the decreased SASA and increased compaction of the aa-tRNA (Fig. 7A). Increased crowding of the aa-tRNA accommodation corridor is consistent with subunit rolling occurring prior to aa-tRNA accommodation, compacting the A site. Increased crowding appears to increase the steric contribution of H89 during accommodation, making it a larger barrier to overcome during accommodation. Surpassing this barrier is aided by the pivoting of the aa-tRNA (up to ~30°) that aligns the elbow domain minor groove and acceptor stem major groove of the tRNA properly to accommodate into the A site (Fig. 7B). This increased barrier that H89 provides is

**Table 1 | Conservation of basic amino acids of domain III of eEF1A that interact with the elbow of the aa-tRNA**

| Amino Acid | Conservation | Other possible identities |
|---|---|---|
| K378 | 94.84% | R, T |
| K392 | 58.71% | A, G, S, H, D, I, P, V, E, Q |
| K395 | 96.13% | A, Q, R |
| R381 | 58.44% | A, P, K |
| R382 | 78.57% | A, S, K |

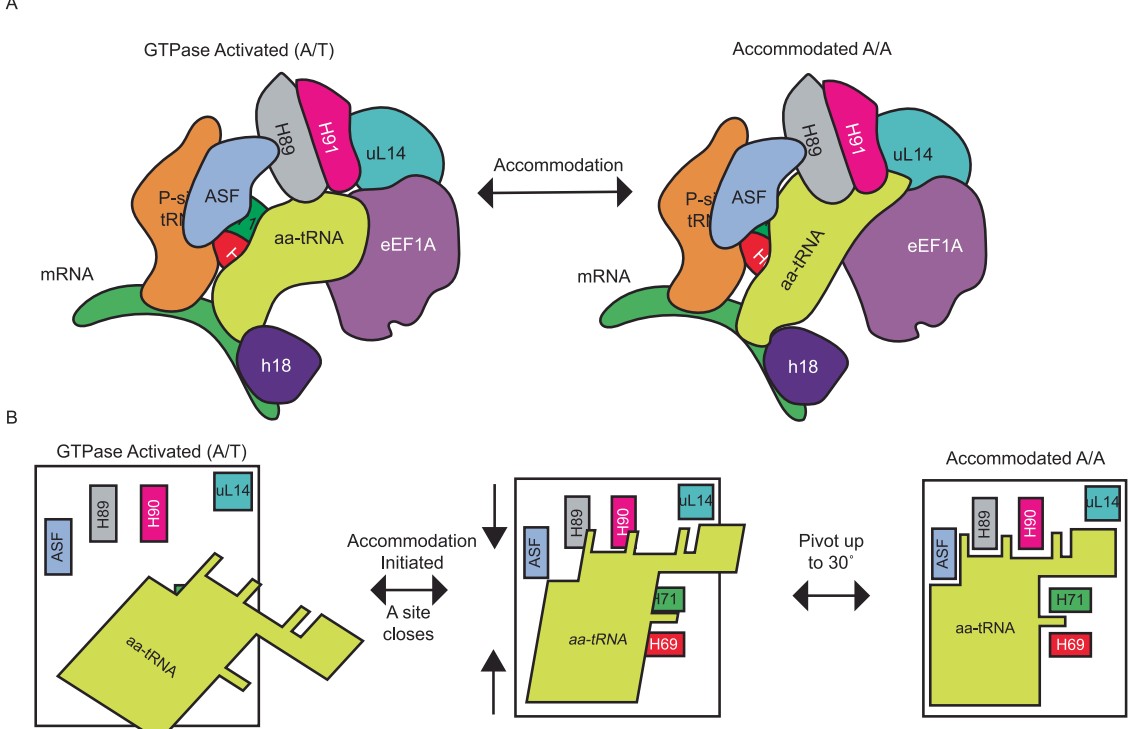

**Fig. 7 | Model of aa-tRNA pivoting into the ribosomal A-site. A** Crowded environment of the human accommodation corridor enables aa-tRNA accommodation to be 10-fold slower. Additional eEF1A stays in complex with the ribosome long enough form transient interactions between domain III of eEF1A and the elbow domain of the accommodating aa-tRNA. **B** aa-tRNA in human approaches the accommodation corridor until it reaches the H89 steric barrier. It then requires pivoting of the aa-tRNA to properly align with all of the elements of the accommodation corridor to enter the A site.

consistent with the tenfold reduced rate in aa-tRNA accommodation into the A site as observed with smFRET studies[68].

In the studies that identified the tenfold reduced rate in aa-tRNA accommodation, subunit rolling was also identified between the GA and AC states of the ribosome[68,69]. It appears that the subunit rolling that occurs during this transition is responsible for the increased barrier of aa-tRNA accommodation, namely by H89. The aa-tRNA can overcome this barrier by swiveling to align the aa-tRNA properly with all of the elements of the accommodation corridor (Fig. 7B). Simultaneously, the accommodating aa-tRNA can fit through a narrower window to accommodate into the A-site of the ribosome. These findings suggest that the ribosome has evolved the accommodation corridor to provide an optimally crowded environment for cognate aa-tRNA selection, similar to how molecular crowding influences the thermodynamics and kinetics of RNA and protein folding[94,95].

Furthermore, the requirement of the aa-tRNA pivoting provides further support for the hypothesis that the geometric alignment of the aa-tRNA relative to catalytic centers of the ribosome is a critical component in maintaining the fidelity of translation[90,96]. In bacteria, we observed that near-cognate aa-tRNA is misaligned relative to catalytic centers in comparison to cognate aa-tRNA. If the same misalignment of near-cognate aa-tRNA occurs in humans, it would be less likely to pivot correctly to overcome the increased barrier of H89. Therefore, the misalignment of the near-cognate aa-tRNA would not be able to align with the accommodation corridor correctly, contributing to the increased accuracy of decoding observed in eukaryotes[8].

The second distinction we observed in aa-tRNA accommodation is that in humans, switch I of eEF1A only interacts with 3′-CCA end minor groove of the aa-tRNA and that domain III of eEF1A made interactions through conserved lysine amino acids with the elbow domain of the aa-tRNA (Fig. 6). In bacteria, it is posited that either the switch I passes through the 3′-CCA end, or the acceptor stem groove to align the aa-tRNA during accommodation into the A site[76]. As human aa-tRNA selectively passes through the 3′-CCA end groove, it indicates that there is a more restrictive selection on the initial trajectory of the aa-tRNA into the ribosomal A site.

The direct interactions observed between domain III of eEF1A through conserved basic amino acids provides a structural framework for how eEF1A can contribute to aa-tRNA proofreading. It was observed that EF-Tu can lower to the barrier of aa-tRNA proofreading by providing steric contributions to aa-tRNA movements towards the A site[77]. Similarly, domain III of eEF1A could provide these steric barriers after domain rearrangement of eEF1A. Together, the contributions of eEF1A and aa-tRNA pivoting provide distinct structural features for how aa-tRNA accommodates into the ribosome compared to bacteria, providing a rationale for the higher accuracy observed in eukaryotes.

## Methods

### Generation of ribosome models with A/T and A/A aa-tRNA

Both the GA and AC complexes with A/T and A/A aa-tRNA positions were modeled prior to the availability of the complexes[68,69]. These states were modeled using the coordinates of the ribosome from PDB ID: 6Y0G. Coordinates of the eEF1A(GTP) phe-tRNA^phe ternary complex bound to ribosomes were derived from PDB ID: 5LZS, a *Oryctolagus cuniculus* ribosomes bound to a compact post-hydrolysis eEF1A in complex with GDP, stalled in a conformation consistent with the GTP bound form by didemnin B[97]. A homology model of the *H. sapiens* eEF1A was generated using the SWISS-MODEL server, using a previously described approach and the GDP was manually converted to GTP in Visual Molecular Dynamics (VMD)[98–101]. Similarly, the tRNA was converted to the sequence of tRNA^phe using the Swapna package in Chimera[102]. Phenylalanine was attached to the 3′−adenosine of the aa-tRNA according to the specific chemistry. The A/A positioned aa-tRNA from the 6Y0G PDB was removed to model the A/T conformation and

replaced with the *H. sapiens* homology model eEF1A(GTP) phe-tRNA^phe ternary complex, representing the GA complex.

To model GDP conformation of eEF1A for the A/A aa-tRNA state the structure of eEF1A in the open, or GDP bound, conformation was used PDB ID: 4C0S[103]. A homology model of the eEF1A(GDP) conformation was made in the SWISS-MODEL server and was aligned to the ribosome by the P-loop of the eEF1A(GTP) model. The eEF1A(GDP) was then minimized in GROMACS 2021 to ensure that there were no overlapping atoms between eEF1A and the ribosome[104–107].

### Model preparation of ribosome complexes

Models of the GA and AC complexes were subject to explicit solvent minimization and equilibration in GROMACS 2021[104–107]. Initially, periodic boundaries extending 15 Å beyond the boundaries of the ribosome complex was filled with SPC/E water molecules and all explicit solvent simulation used the AMBER 99 forcefield. Ions were added at a ratio of 20 mM Mg and 200 mM K per water molecule and sufficient Cl to neutralize the system, to maintain ionic strength conditions similar to previous ribosome simulations[74,91]. Water was removed and the ions were simulated with a frozen ribosome, in implicit water conditions with a dielectric of 80, to allow for Mg and K to condense on the ribosome using modified ion parameters to limit ion-ion interactions[108]. The Van der Waals radii of the ions were increased to mimic a hydrated ion so that they would properly condense on the ribosome. SPC/E water was added back to the system and the water was minimized using a steepest decent approach followed by minimization of the ions, with their Van der Waals radii returned. An NVT equilibration at 300 K of the ions and solvent was performed for 10 ns. Subsequently, an NPT equilibration was performed at 300 K and 1 atmosphere was performed for 10 ns while slowly releasing restraints on the ribosome.

### Structure-based simulations of ribosome complexes

Ribosome coordinates after 20 ns of equilibration were used as starting structures for structure-based simulations of the aa-tRNA proofreading process using SMOG 2.4 source code[109]. Two different potentials were implemented in this manuscript defined as potential 1 and potential 2. These two potentials differ in how they define the contacts that define the energetic minima in the structure-based simulations.

Potential 1 ($V_1$) used in the structure-based simulations is defined as:

$$V_1 = \sum_{bonds} \frac{\varepsilon_r}{2}(r_i - r_{i,o})^2 + \sum_{angles} \frac{\varepsilon_\theta}{2}(\theta_i - \theta_{i,o})^2$$
$$+ \sum_{impropers} \frac{\varepsilon_{\chi i}}{2}(X_i - X_{i,o})^2 + \sum_{planar} \frac{\varepsilon_{\chi p}}{2}(X_i - X_{i,o})^2$$
$$+ \sum_{backbone} \varepsilon_{BB} F_D(\phi_i - \phi_{i,o}) + \sum_{sidechains} \varepsilon_{SC} F_D(\phi_i - \phi_{i,o}) \quad (1)$$
$$+ \sum_{contacts} \varepsilon_C \left[ \left(\frac{\sigma_{ij}}{r_{ij}}\right)^{12} - 2\left(\frac{\sigma_{ij}}{r_{ij}}\right)^6 \right] + \sum_{non-contacts} \varepsilon_{NC} \left(\frac{\sigma_{NC}}{r_{ij}}\right)^{12}$$

where,

$$\varepsilon F_D(\phi) = \varepsilon(1 - \cos \phi) + \frac{\varepsilon}{2}(1 - \cos 3\phi) \quad (2)$$

Where $\varepsilon_r = 50\,\varepsilon_0$, $\varepsilon_\theta = 40\,\varepsilon_0$, $\varepsilon_{\chi i} = 10\,\varepsilon_0$, $\varepsilon_{\chi p} = 40\,\varepsilon_0$, $\varepsilon_{NC} = 0.1\,\varepsilon_0$, $\sigma_{NC} = 2.5\,\text{Å}$, and $\varepsilon_0 = 1$. Native contacts were defined as atom pairs within 4.5 Å of each other. This potential was implemented to capture the reversible fluctuations of the aa-tRNA during proofreading. In these simulations the contacts of the aa-tRNA in the A/A tRNA position were scaled by a factor of 0.13 and contacts between the codon and anticodon were scaled by a factor of 0.8. This scaling was implemented to achieve reversible fluctuations of the aa-tRNA and to model a system

where the mRNA and aa-tRNA interactions are reversible. Using this model the aa-tRNA rapidly reaches the elbow accommodated-1 (EA-1) position and rapidly fluctuates between the EA-1 and elbow accommodated-2 position (EA-2). This potential capture reversible nature of accommodation but did not capture early and late-stage events during aa-tRNA accommodation. Each system was simulated for a total of $1 \times 10^6 \, \tau_{ru}$ (time reduced units) or $5 \times 10^8$ timesteps with a timestep of $0.002 \, \tau_{ru}$ using GROMACS 2021[104–107]. Simulations were performed at a constant temperature of 0.5 reduced units. Within these simulations we captured 1769 aa-tRNA accommodation into the A site events. Accommodation events were considered when the elbow domain of the aa-tRNA reached a distance of <32.5 Å, indicating an elbow accommodated position[77].

Potential 2 (V$_2$) was similarly defined, however, the contact terms were replaced with a single Gaussian term (Supplementary Methods)[90]. Similarly, the contacts of the final accommodated state (AC) between the aa-tRNA and the ribosome were scaled by 0.4 and between the mRNA and tRNA by 0.8. The advantage of these simulations is they enable the identification of early and late-stage accommodation events. However, they do not capture the reversible fluctuations of the aa-tRNA during accommodation. Simulations using potential 2 also enabled the comparison between aa-tRNA proofreading of prokaryotic (*E. coli*) and eukaryotic (*H. sapiens*) ribosomes (Supplementary Methods). Each simulation was performed for at least $1 \times 10^7$ timesteps using a timestep is $0.002 \, \tau_{ru}$. Using this simulation setup we performed 100 simulations. Of these 100 simulations we observed 87 elbow accommodation events with 13 aa-tRNA dissociating prior to accommodation (Supplementary Table 3). For simulations without H89 or H69/H71 we used potential 2, deleting each helix[77].

### Simulations of prokaryotic ribosomes
Prokaryotic ribosomes were prepared as previously described[90]. Potential 2 was used to characterize early- and late-stage accommodation events. Each simulation was performed for $1.25 \times 10^7$ timesteps using a timestep is $0.002 \, \tau_{ru}$.

### Distance measurements
To measure the progression of aa-tRNA during accommodation into the A site, we measured the distance between the O3' oxygen of the U60 and U8 nucleotides of the aa-tRNA and peptidyl-tRNA respectively ($R_{elbow}$). This distance was used to describe the movement of the tRNA elbow domain into the A site of the ribosome, similar to $R_{elbow}$ measured for accommodating tRNA in *E. coli*[90]. Similarly, we measured the distance between A76 of the aa-tRNA and peptidyl tRNA to describe late-stage accommodation events of aa-tRNA, similar to $R_{CCA}$ as previously described in *E. coli*[90]. Subunit rolling ($R_{rolling}$) was measured using the distance between the O3' atoms of A465 of h14 and U4559 of the SRL. To describe the movement of switch I relative to domain III of eEF1A we measured the distance of the Cα atoms of R69 and A425 ($R_{swi-DIII}$). To quantify the movements of switch I relative to the 3'-CCA end of the tRNA during accommodation we measured the distance between the Cα atom R69 of eEF1A and the O3' atom of A76 of the accommodating aa-tRNA ($R_{swi-CCA}$). Lastly, to describe the position of domain III of eEF1A relative to the elbow domain of aa-tRNA we measured the distance of the Cα atom of K378 to U55 O3' ($R_{swi-elbow}$). Average distance measurements were determined using a single gaussian distribution fit as defined by:

$$counts = Ae^{\frac{-(x-\bar{x}_a)^2}{2s_a^2}} \qquad (3)$$

where A is the peak of the distribution, $\bar{x}_a$ is the average value of the distance, and s$_a$ is the standard deviation for the population.

### tRNA angle measurement
Measurements of the tRNA angle ($\theta_{tRNA}$) during accommodation were performed in VMD. Here, a plane was generated across the accommodating tRNA from the O3' atoms of C4, A35, and G56. A vector perpendicular to the plane of the accommodating tRNA is generated and the change in the angle of the vector throughout the simulation is measured, using the first frame as a reference.

### Approximate free energy landscape
All free energy landscape approximations were generated using two reaction coordinates (i.e., $R_{elbow}$ and $\theta_{tRNA}$) and were compared in a Boltzmann weighted approximated free energy landscape using Eq. 4[90].

$$\Delta G^* = -k_B T \ln\left(\frac{P(x_i)}{P_{max}(x)}\right)^2 \qquad (4)$$

Here, ΔG* is the relative free energy of the landscape, $K_B$ is the Boltzmann coefficient, T is the temperature (300 K), P(x$_i$) is the probability density function of being at state i obtained from a histogram of the compared measurements, and $P_{max}(x)$ is the maximum probability of the most observed state.

### Convergence of simulations
Convergence of the structure-based simulations was calculated using the pointwise RMSD between successive free energy landscapes as previously described[90,91]. The pointwise RMSD convergence metric is defined by Eq. 5:

$$Conv(t) = \sqrt{\sum_{i,j}\frac{\sum_{i,j}\left(\Delta G^*(i,j)_t - \Delta G^*(i,j)_{t0}\right)^2}{N}} \qquad (5)$$

where ΔG*(i,j)$_{t0}$ is the approximate free energy landscape of $R_{elbow}$ and $\theta_{tRNA}$ after sampling to time t$_0$ (i.e., 1000 frames of simulation) and ΔG*(i,j)$_t$ is the approximate free energy landscape of $R_{elbow}$ and $\theta_{tRNA}$ after sampling to time t, and N is the number of grid points on the free energy landscape. After Conv(t) plateaus, simulations were considered converged.

### Reporting summary
Further information on research design is available in the Nature Portfolio Reporting Summary linked to this article.

## Data availability
Structures, parameters, and example trajectories for this study have been deposited in the Zenodo database [https://doi.org/10.5281/zenodo.15653228]. This includes the initial PDB file used, the input parameters for gromacs simulations (i.e., gro, top, and mdp files), as well as the tpr files. Information on the MD simulations is provided (Supplementary Table 4). Source data is provided with this paper. Source data are provided with this paper.

## Code availability
All scripts that were developed for the analysis of this project have been uploaded to Zenodo [https://doi.org/10.5281/zenodo.15653228].

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

## Acknowledgements
The authors would like to thank all members of the Girodat and Sanbonmatsu labs for their helpful comments and discussions. The authors would also like to thank LANL Institutional Computing and Digital Research Alliance of Canada for their generous support. This work was supported by NIGMS R15-GM151696-01, University of Lethbridge startup funds, and NSERC Discovery Grant (RGPIN-2025-04896) (to D.G.), and NIH NIGMS 1R35GM156585 (to K.Y.S.).

## Author contributions
D.G. designed the experiments, performed the simulations and subsequent analysis. D.S. performed the sequence alignment and all proteomic analysis. D.G. and K.Y.S. contributed to the writing of the manuscript.

## Competing interests
The authors declare no competing interests.
