## [Transparent Peer Review file · Nature Communications]

Human Protein Synthesis Requires Aminoacyl-tRNA Pivoting During Proofreading

Corresponding Author: Dr Dylan Girodat

Version 0:

Reviewer comments:

Reviewer #1

(Remarks to the Author)

This manuscript presents a study of aa-tRNA accommodation through structure-based molecular simulation. The authors employed two potentials to capture the reversible fluctuations of aa-tRNA and the early and late stages of the accommodation events. They identified a distinct pivoting of the aa-tRNA during accommodation in humans, which is attributed to the increased contact surface between tRNA and the corridor, consistent with the subunit rolling that occurs prior to accommodation. This finding provides an explanation for the observation that tRNA selection is slower in humans compared to *E. coli*. Furthermore, the authors identified the crucial role of eEF1A in aa-tRNA accommodation in humans, as it interacts with the 3'-CCA end minor groove of the aa-tRNA, with its domain III engaging in interactions through conserved residues.

My comments are listed below.

Major

Line 108. How is the "accommodation event" defined in the simulation? Is it determined when the simulation reaches convergence? Are there instances of failed events in the simulation that do not achieve convergence?

It appears that the authors also conducted simulations for bacterial aa-tRNA selection in *E. coli* (line 80). Do the simulations for bacteria also exhibit pivoting and fluctuations similar to those observed in humans? Do the two conformations, EA1 and EA2, exist in the bacteria? More detailed comparisons are necessary to highlight the differences between aa-tRNA selection in humans and bacteria.

A key finding of this study is that the steric barrier imposed by H89 is more significant in humans than in bacteria, which can explain the tenfold reduction in the rate of tRNA selection. Can this comparison be made more quantitatively? Can the authors estimate the aa-tRNA accommodation rate in humans based on the simulation data? Is it approximately tenfold that of bacteria?

Line 269: The authors studied the conservation of residues in Domain II of eEF1A that interact with aa-tRNA. Do the nucleotides in the aa-tRNA elbow also show conserved features?

Line 381. How were the scaling factors (0.13 and 0.8 in potential 1, 0.4 and 0.8 in potential 2) selected?

Minor

Line 48-51. Citations are needed for the methodologies mentioned.

Line 78. Two "tRNA selection" are mentioned in the sentence.

Line 96. What is the meaning of "native contacts" and "non-native contacts"? A clearer definition is necessary.

Line 369. What's the full name of "SBM"?

Line 307. Incomplete sentence "This".

Line 475. (B) should be (D).

Line 488. More description for Panel B is needed. How are the three cartoon representations related?

Reviewer #2

(Remarks to the Author)

Experimental high- and low-resolution methods allow the identification of structural states that are key to modeling dynamic mechanisms such as protein decoding by the ribosome and transfer RNAs (tRNAs). Models of decoding have, for example, captured important differences between bacterial and eukaryotic translation. However, the experimentally captured states do not elucidate the dynamic transitions between them. This limitation can be seen as a current boundary of experimental techniques. In this study, the authors use molecular dynamics simulations to model the transition dynamics of an aminoacyl-tRNA (aa-tRNA) accommodating into the A-site of the human ribosome. Their simulations reveal a crucial $\sim 30^\circ$ pivoting of the anticodon stem within the accommodation corridor. This pivoting appears essential for navigating a more crowded pathway in eukaryotes, ensuring faithful translation. The simulations suggest a decoding dynamic that is distinct in eukaryotes compared to prokaryotes.

The manuscript is clearly written and easy to follow. The conclusions drawn by the authors are well supported by their simulation results. The methods used represent the current state of the art in RNA molecular dynamics. The figures are particularly helpful for visualizing the simulation outcomes and understanding the observed molecular behavior. The proposed model is clearly illustrated in Figure 7B.

Experimental validation will be necessary to confirm the theoretical results. This reflects a broader trend as experimental methods approach their resolution limits. The results provide a valuable framework for understanding the biological phenomenon under study, specifically, the structural dynamics of the ribosome's A-site and its interaction with aa-tRNA. These findings fit logically into the ongoing structural and dynamic studies of the ribosome by this research group. The proposed framework will also support the formulation of new hypotheses and the design of novel experiments that may, in turn, yield further insights into decoding in humans and eukaryotes.

Line 307: The word "This" appears mistakenly at the end of the paragraph. Either a full sentence was accidentally deleted, or "This" was left behind from a previous edit.

Reviewer #3

(Remarks to the Author)

Reviewer #4

(Remarks to the Author)

The manuscript shows valuable computational insights into the mechanism of aa-tRNA pivoting within the accommodation corridor in the human ribosome. Overall, this is a very good, high-quality work, of interest for a wide scientific community.

One significant limitation of this paper is the lack of accessible data, which prevents the visualization and inspection of the structures and trajectories. Since the input files for the simulations with the full set of parameters are also not available, this work cannot be reproduced. In modern research practices, it is both feasible and common to deposit data in repositories. While the full trajectories generated in this study are too large to upload, a reduced version — such as a trajectory stripped of water molecules and ions, comprising only several dozen frames — should be shared.

Even though the authors attempted to demonstrate the convergence of the simulations in Figure S1, I do have doubts that the plateau is reached, especially in the case of simulations where native contacts are defined using potential 1. Figure S1 rather demonstrates that these simulations should be extended. Moreover, this may be the reason why the 3'-CCA end rarely reached its final accommodated position in the simulations with potential 1.

Figure 2 shows "representative distances". It would be beneficial to visualize the distances in all simulations within the Supporting Information, rather than restricting the visualization to those in just two arbitrarily selected simulations. The final 20 μ s of the simulation depicted in Figure 2E,F exhibit a distinct deviation from the pattern observed in the other fluctuations, further highlighting the issue of insufficient simulation duration.

The overall quality of the figures could be improved. In all main text figures, the all-atom structural representations of the ribosome lack sufficient contrast. Due to the overlay of the captions and the structure, and very thin text outline, the captions

become difficult to read. Perhaps the text outline is not necessary or it could be adjusted to the minimum line width for print as it is currently too thin to be visualized properly at this image resolution. Additionally, in Figure 1, the captions could be placed next to one schematic picture only, there is no need to repeat them in every panel as they are identical. In Figure 3, it is very difficult to notice the aa-tRNA contacts that are described in the text. Panels C and D look almost the same as there are no insets, no highlight of differences. In Figure 3, some fonts are so small that it is impossible to read the labels. Looking at Figure 4, the definition of theta_tRNA is missing, and it is unclear how the 30-degree angle is calculated or how it remains consistent across two different perspectives. Does this angle correspond to theta_tRNA? The lack of clarity in both the labels and the figure caption makes it confusing.

Given that this manuscript highlights the distinctions between human and prokaryotic ribosomes, creating a comparable movie focusing on the prokaryotic ribosome (or adding a fragment to the current movie) could enhance the understanding of these differences. In the movie, it would be good to label the 3' CCA end of aa-tRNA.

From a technical perspective, I noticed spelling errors, such as "respectively" (page 3, line 9) and "investagations" (page 4, line 4), and some mistakes such as "if whether" (page 10, line 11), "This " (page 14, line 20). I find it distracting to use R_CCA and R_cca interchangeably. In my opinion, R_CCA is more appropriate, taking into account the meaning.

Version 1:

Reviewer comments:

Reviewer #1

(Remarks to the Author)

The authors have satisfactorily addressed the original comments of this reviewer.

Reviewer #3

(Remarks to the Author)

Reviewer #4

(Remarks to the Author)

The manuscript has improved significantly and all my concerns have been addressed. I believe that the minor errors, like spelling mistakes and the existence of R_CCA and R_cca used interchangeably in some figures, will be corrected at the proofing stage.

REVIEWER COMMENTS

Reviewer #1 (Remarks to the Author):

This manuscript presents a study of aa-tRNA accommodation through structure-based molecular simulation. The authors employed two potentials to capture the reversible fluctuations of aa-tRNA and the early and late stages of the accommodation events. They identified a distinct pivoting of the aa-tRNA during accommodation in humans, which is attributed to the increased contact surface between tRNA and the corridor, consistent with the subunit rolling that occurs prior to accommodation. This finding provides an explanation for the observation that tRNA selection is slower in humans compared to *E. coli*. Furthermore, the authors identified the crucial role of eEF1A in aa-tRNA accommodation in humans, as it interacts with the 3'-CCA end minor groove of the aa-tRNA, with its domain III engaging in interactions through conserved residues.

My comments are listed below.

Major

Line 108. How is the "accommodation event" defined in the simulation? Is it determined when the simulation reaches convergence? Are there instances of failed events in the simulation that do not achieve convergence?

We thank the reviewers for the opportunity to clarify these concepts in the manuscript. To improve clarity, we have described how accommodation events are determined by adding:

"Accommodation events were considered when the elbow domain of the aa-tRNA reached a distance of $< 32.5 \text{ \AA}$, indicating an elbow accommodated position⁵⁴" to the methods section.

We have also added clarification on the number of failed accommodation events that do occur in our simulations. Although no failed accommodation events are observed using potential 1 we observed that 13 of our potential 2 simulations do not reach the accommodated state and the aa-tRNA dissociates. We have clarified this in the manuscript by adding.

"In using potential 2 we observed in 13 simulations where the aa-tRNA dissociates prior to accommodation, representing rejected aa-tRNA." To the results

and

"Using this simulation setup we performed 100 simulations. Of these 100 simulations we observed 87 accommodation events with 13 aa-tRNA dissociating prior to accommodation." To the methods section.

It appears that the authors also conducted simulations for bacterial aa-tRNA selection in *E. coli* (line 80). Do the simulations for bacteria also exhibit pivoting and fluctuations similar to those observed in humans? Do the two conformations, EA1 and EA2, exist in the bacteria? More detailed comparisons are necessary to highlight the differences between aa-tRNA selection in humans and bacteria.

This is a great point and we are grateful for the opportunity to clarify this in the manuscript. Yes, the EA1 and EA2 positions are observed in bacteria and have been described by cryo-electron microscopy previously in Loveland et al. Nature 2020. We have added to Supplemental Figure 4 to show the EA-1 and EA-2 positions that are observed in our simulations. To clarify this we have added:

*“These positions do reflect the elbow-accommodation aa-tRNA position that has been observed in *E. coli* previously⁵⁰; therefore, EA-1 and EA-2 are reflective of tRNA positions that are on-path towards the A/A position but not completely there. Both the EA-1 and EA-2 positions were observed in our simulations of bacterial accommodation (**Supplemental Fig. 4**). To the results section.*

tRNA pivoting is also not observed in bacteria as highlighted in Supplemental Figure 5. We also added the text.

*“No pivoting was observed in our bacterial simulations (**Supplemental Fig. 5A**).” to our results section.*

A key finding of this study is that the steric barrier imposed by H89 is more significant in humans than in bacteria, which can explain the tenfold reduction in the rate of tRNA selection. Can this comparison be made more quantitatively? Can the authors estimate the aa-tRNA accommodation rate in humans based on the simulation data? Is it approximately tenfold that of bacteria?

This is a great idea from the reviewer and one that we considered when writing the manuscript. When it comes to the structure-based models the accommodation events correlate strongly with the attempt frequency of accommodation and are not a direct reflection of the rate. However, one may use this attempt frequency to estimate the rate of the reaction from these simulations if the activation energy is known (E_a). As E_a is not clear for human aa-tRNA accommodation we did not directly perform a comparison. What we can do is estimate the E_a based on the known rates of bacterial and human accommodation and use the barrier-crossing attempt frequency calculated from simulations using the equation $k = A e^{-\frac{E_a}{k_B T}}$, where k is the rate, A is the barrier-crossing attempt frequency, k_B is the Boltzmann constant and T is temperature. We have done this and found that the E_a estimated for bacterial accommodation matches previously determined values and that the E_a for tRNA accommodation in humans is 11-13.4 $k_B T$. We have added a table summarizing this in the supplemental and have included the comparison in the main text by adding:

“By using the number of timesteps required for each accommodation event in our simulations we could estimate the barrier height during transition from the A/T position

to the transition state using previously determined rates of accommodation (**Supplemental Methods**). From these measurements we estimate that the barrier height of *E. coli* accommodation is 9.3 – 10.4 $k_B T$, similar to previously determined values, while in humans the barrier is 11-13.4 $k_B T$ ⁸⁰.”

Line 269: The authors studied the conservation of residues in Domain II of eEF1A that interact with aa-tRNA. Do the nucleotides in the aa-tRNA elbow also show conserved features?

*We thank the reviewers for the opportunity to include a description of the nucleotide conservation of the tRNA in our manuscript. We performed a sequence alignment of tRNA in humans and tRNA from various species (*H. sapiens*, *S. cerevisiae*, *M. musculus*, *B. subtilis*, *E. coli*, *D. melanogaster*, *A. thaliana*, and *S. pombe*) to identify the conservation of the nucleotides that interact with domain III. The sequence conservation varies from 0.43% to 99.31%, indicating that the interactions between the tRNA and eEF1A are based on the tRNA structure rather than sequence. This is supported by all of the interactions being mediated by the phosphodiester backbone of the tRNA rather than the nitrogenous base. To help describe this we have added:*

*“The conservation of nucleotides of the tRNA that interact with domain III of eEF1A were also considered, either amongst humans or throughout different domains of life (G1, U51, G52, A64, G65, and U66) (**Supplemental Table 2**). Although the conservation of the nucleotides varied (0.43% to 99.31%), all interactions between the tRNA and domain III of eEF1A were mediated by backbone interactions, indicating that the nucleoside identity is not essential for these interactions (**Figs. 6 D,E**).”* to the results

And included a supplemental table describing the conservation of the nucleotides in tRNA.

Line 381. How were the scaling factors (0.13 and 0.8 in potential 1, 0.4 and 0.8 in potential 2) selected?

To help clarify why these scaling factors were chosen we have included a further description to paragraph 2 of the "Simulations of aa-tRNA proofreading in humans" section. This paragraph now reads:

*“To achieve reversible fluctuations of the aa-tRNA in the simulations with potential 1 we scaled the weight of the contacts for the tRNA in the A/A position in the ribosome. By titrating the weight of the native contacts for the A/A position of the tRNA we were able to fine-tune our simulations to be evenly distributed between the two elbow accommodated positions (EA-1 and EA-2) in the simulations at a contact weight of 0.13 (**Supplemental Fig. 2**). Similarly, the contacts for the A/A position in potential 2 simulations were scaled by 0.4 to enable fluctuations of the aa-tRNA during accommodation, as previously performed in bacterial systems⁹¹. A rescaling of 0.4 enables the tRNA to accommodate into the A-site through reversible fluctuations, reflecting tRNA trajectories seen in smFRET studies^{35,68}. The interactions between the mRNA and tRNA were scaled by a factor of 0.8 to compare to bacterial studies where it was rescaled by this value to reflect tRNA rejection frequencies⁹¹.”*

Minor

Line 48-51. Citations are needed for the methodologies mentioned.

We have included the following citations to the sentence:

This mechanism has been extensively biochemically characterized and structurally validated through various methodologies including single molecule Fluorescence (Förster) Resonance Energy Transfer (smFRET)²⁰⁻³⁰, cryo-electron microscopy (cryo-EM)³¹⁻³⁹, X-ray crystallography^{25,29,40-59}, and pre-steady state kinetics⁶⁰⁻⁶³.

Line 78. Two “tRNA selection” are mentioned in the sentence .

Thank you, this has been corrected to read

“Moreover, they have described the roles of EF-Tu^{29,44,54,67,81}, the A-site finger (helix 38 of the 23S rRNA)⁵³, L11 flexibility⁸², tRNA selection⁸³, and tRNA diffusion⁸⁴, as well as ribosome-stimulated EF-Tu GTP hydrolysis⁸⁵⁻⁹⁰.”

Line 96. What is the meaning of "native contacts" and “non-native contacts”? A clearer definition is necessary.

Native contacts refers to the atoms that are within the cutoff distance in the fully accommodated A/A tRNA conformation. Non-native contacts refers to interactions between atoms that are not within the cutoff distance of the fully accommodated A/A tRNA conformation. We have included the definition of:

“Native contacts are defined as atom pairs within 4.5 Å of each other in the ribosome when the aa-tRNA is in the accommodated A/A conformation, while non-native contacts refer to interactions not present in this conformation.”

Line 369. What’s the full name of “SBM”?

SBM stands for structure-based models. This has been removed from the text and replaced with “structure-based simulations”

Line 307. Incomplete sentence “This”.

We have removed it from the manuscript to improve clarity.

Line 475. (B) should be (D).

We appreciate the reviewer for pointing out the typographical error. We have edited (B) to (D) in the text.

Original sentence: (B) Representative structure of EA-2.

Edited sentence: (D) Representative structure of EA-2.

Line 488. More description for Panel B is needed. How are the three cartoon

representations related?

Thank you for the opportunity to clarify we have added to the figure caption:

“(B) Cartoon representations of the aa-tRNA pivoting into the ribosomal A site. aa-tRNA begins accommodation into the ribosome in the bent non-pivoted position (left), the tRNA then needs to pivot and accommodate through a 3'CCA-end first mechanism (middle), then the tRNA returns to a non-pivoted position once the elbow of the tRNA enters the A site.”

Reviewer #2 (Remarks to the Author):

Experimental high- and low-resolution methods allow the identification of structural states that are key to modeling dynamic mechanisms such as protein decoding by the ribosome and transfer RNAs (tRNAs). Models of decoding have, for example, captured important differences between bacterial and eukaryotic translation. However, the experimentally captured states do not elucidate the dynamic transitions between them. This limitation can be seen as a current boundary of experimental techniques. In this study, the authors use molecular dynamics simulations to model the transition dynamics of an aminoacyl-tRNA (aa-tRNA) accommodating into the A-site of the human ribosome. Their simulations reveal a crucial $\sim 30^\circ$ pivoting of the anticodon stem within the accommodation corridor. This pivoting appears essential for navigating a more crowded pathway in eukaryotes, ensuring faithful translation. The simulations suggest a decoding dynamic that is distinct in eukaryotes compared to prokaryotes.

The manuscript is clearly written and easy to follow. The conclusions drawn by the authors are well supported by their simulation results. The methods used represent the current state of the art in RNA molecular dynamics. The figures are particularly helpful for visualizing the simulation outcomes and understanding the observed molecular behavior. The proposed model is clearly illustrated in Figure 7B.

Experimental validation will be necessary to confirm the theoretical results. This reflects a broader trend as experimental methods approach their resolution limits. The results provide a valuable framework for understanding the biological phenomenon under study, specifically, the structural dynamics of the ribosome's A-site and its interaction with aa-tRNA. These findings fit logically into the ongoing structural and dynamic studies of the ribosome by this research group. The proposed framework will also support the formulation of new hypotheses and the design of novel experiments that may, in turn, yield further insights into decoding in humans and eukaryotes.

Line 307: The word “This” appears mistakenly at the end of the paragraph. Either a full sentence was accidentally deleted, or “This” was left behind from a previous edit.

We appreciate the reviewer pointing out incomplete text. We have removed it from the manuscript to improve clarity.

Reviewer #3 (Remarks to the Author):

Reviewer #4 (Remarks to the Author):

The manuscript shows valuable computational insights into the mechanism of aa-tRNA pivoting within the accommodation corridor in the human ribosome. Overall, this is a very good, high-quality work, of interest for a wide scientific community.

One significant limitation of this paper is the lack of accessible data, which prevents the visualization and inspection of the structures and trajectories. Since the input files for the simulations with the full set of parameters are also not available, this work cannot be reproduced. In modern research practices, it is both feasible and common to deposit data in repositories. While the full trajectories generated in this study are too large to upload, a reduced version — such as a trajectory stripped of water molecules and ions, comprising only several dozen frames — should be shared.

We have uploaded the files necessary for simulations (i.e. PDB, GRO, TOP) files as well as example trajectories and example script files to Zenodo. These files are restricted but will be made available upon publication. For reviewers we can provide access.

The doi is [10.5281/zenodo.15653228](https://doi.org/10.5281/zenodo.15653228)

Even though the authors attempted to demonstrate the convergence of the simulations in Figure S1, I do have doubts that the plateau is reached, especially in the case of simulations where native contacts are defined using potential 1. Figure S1 rather demonstrates that these simulations should be extended. Moreover, this may be the reason why the 3'-CCA end rarely reached its final accommodated position in the simulations with potential 1.

We thank the reviewer for their comments on the convergence of the simulations. We have extended simulations to 1.5 ms to demonstrate that at this timescale a plateau is maintained, which is comparable to the values at 1 ms. The pointwise RMSD measurement of free energy landscape that we have performed is a highly stringent measure of convergence. When we use a more typical measurement of convergence such as backbone RMSD a plateau is clearly reached in the beginning of the simulation. To demonstrate this, we have included the backbone RMSD to demonstrate convergence of the simulation while including the stringent measurements of pointwise RMSD of the free energy landscapes at 1 and 1.5ms (Supplemental Figs. 1,2)

Within the text we have clarified this with “The convergence of both types of simulations were defined by the typical measurement of backbone RMSD of the ribosome or aa-tRNA in addition to the pointwise RMSD between successive free energy landscapes R_{elbow} and of θ_{tRNA} , (Supplemental Figs. 1, 2).”

Figure 2 shows "representative distances". It would be beneficial to visualize the distances in all simulations within the Supporting Information, rather than restricting the visualization to those in just two arbitrarily selected simulations. The final 20 μs of the simulation depicted in Figure 2E,F exhibit a distinct deviation from the pattern observed in the other fluctuations, further highlighting the issue of insufficient simulation duration.

Additional supplemental figures for the R_{elbow} and R_{cca} measurements from all simulations have been included as supplemental figures 3, 4, 6, and 7. We have also extended these figures to show from 0 to 1ms of simulation time to show that the deviation in the final 20 μs is within the normal range of fluctuations for the system.

The overall quality of the figures could be improved. In all main text figures, the all-atom structural representations of the ribosome lack sufficient contrast. Due to the overlay of the captions and the structure, and very thin text outline, the captions become difficult to read. Perhaps the text outline is not necessary or it could be adjusted to the minimum line width for print as it is currently too thin to be visualized properly at this image resolution. Additionally, in Figure 1, the captions could be placed next to one schematic picture only, there is no need to repeat them in every panel as they are identical. In Figure 3, it is very difficult to notice the aa-tRNA contacts that are described in the text. Panels C and D look almost the same as there are no insets, no highlight of differences. In Figure 3, some fonts are so small that it is impossible to read the labels. Looking at Figure 4, the definition of θ_{tRNA} is missing, and it is unclear how the 30-degree angle is calculated or how it remains consistent across two different perspectives. Does this angle correspond to θ_{tRNA} ? The lack of clarity in both the labels and the figure caption makes it confusing.

We thank the reviewers for their comments on the figures. We have worked to improve the figures by either removing the text outline or making it thinner, where appropriate throughout the manuscript. We have also removed labels for the ribosome structure in figure 1 and left the labels for the cartoon models. We have also changed the full ribosome figures to improve the contrast.

In Figure 4 we have added a definition of θ_{tRNA} and have also referred to the materials and methods and results sections where it is defined. We have clarified that the angle described in panel A is θ_{tRNA} and that it corresponds to the pivoting of the tRNA during accommodation. We have included how θ_{tRNA} is calculated in the methods section.

In Figure 3 we have added zoomed in images of the contacts between the aa-tRNA and regions of the ribosome that are highlighted in the text to enable to reader to accurately

interpret these interactions.

Given that this manuscript highlights the distinctions between human and prokaryotic ribosomes, creating a comparable movie focusing on the prokaryotic ribosome (or adding a fragment to the current movie) could enhance the understanding of these differences. In the movie, it would be good to label the 3' CCA end of aa-tRNA.

We thank the reviewers for the comment and have included a movie of both the eukaryotic and bacterial accommodation trajectories while highlighting the 3'CCA ends of the tRNA.

From a technical perspective, I noticed spelling errors, such as "respectively" (page 3, line 9) and "investagations" (page 4, line 4), and some mistakes such as "if whether" (page 10, line 11), "This " (page 14, line 20). I find it distracting to use R_CCA and R_cca interchangeably. In my opinion, R_CCA is more appropriate, taking into account the meaning.

We thank the reviewer for highlighting the spelling errors. We have corrected these in the manuscript.